# PERSIST platform provides programmable RNA regulation using CRISPR endoRNases

Breanna DiAndreth [1,2], Noreen Wauford[1,2], Eileen Hu[1,2,3,4], Sebastian Palacios[1,2,3,4] & Ron Weiss [1,2,3 ✉]

Regulated transgene expression is an integral component of gene therapies, cell therapies and biomanufacturing. However, transcription factor-based regulation, upon which most applications are based, suffers from complications such as epigenetic silencing that limit expression longevity and reliability. Constitutive transgene transcription paired with post-transcriptional gene regulation could combat silencing, but few such RNA- or protein-level platforms exist. Here we develop an RNA-regulation platform we call "PERSIST" which consists of nine CRISPR-specific endoRNases as RNA-level activators and repressors as well as modular OFF- and ON-switch regulatory motifs. We show that PERSIST-regulated transgenes exhibit strong OFF and ON responses, resist silencing for at least two months, and can be readily layered to construct cascades, logic functions, switches and other sophisticated circuit topologies. The orthogonal, modular and composable nature of this platform as well as the ease in constructing robust and predictable gene circuits promises myriad applications in gene and cell therapies.

[1] Department of Biological Engineering, Massachusetts Institute of Technology, Cambridge, MA 02139, USA. [2] Synthetic Biology Center, Massachusetts Institute of Technology, Cambridge, MA 02139, USA. [3] Department of Electrical Engineering and Computer Science, Massachusetts Institute of Technology, Cambridge, MA 02139, USA. [4] These authors contributed equally: Eileen Hu and Sebastian Palacios. ✉email: rweiss@mit.edu

Transcription factor-based gene regulation in mammalian systems remains a cornerstone of both basic biological research and the field of synthetic biology. Transcription factor-promoter pairs such as the Tet-On or Tet-Off systems[1–3] have been used extensively for transgene modulation while the robust behavior, modularity, composability, and large design space of many other additional transcriptional regulators[4–7] have also enabled their layering into multi-transcription factor circuits for forward-design of complex cellular behavior. However, hurdles such as epigenetic silencing challenge the therapeutic and biomanufacturing potential of these strategies. Epigenetic silencing is the prevention of gene expression that is typically regulated by DNA methylation and chromatin remodeling (e.g., by histone modifications)[8]. Epigenetic silencing of transgenes has been observed in many therapeutically-relevant cell types including stem cells[9], neurons[10], and CHO cells[11], and inhibits the engineered functions of these cells over time. While the location of genomic integration has been shown to be one factor that effects silencing[12–17], other emerging factors such as features of promoter sequences[11,18–22] and preferences for a transcriptionally active state[9,23–26] make transcription factor-based gene regulation unsuitable in many contexts.

Recently developed post-transcriptional platforms comprising activator- and repressor-like regulators may be able to address this silencing problem. Such platforms would allow the use of constitutive promoters routinely employed in gene therapy, which tend to resist silencing[27,28], and may also find use in modern therapeutic modalities such as mRNA gene therapies. Other options are the new protein-based protein-regulation platforms that use orthogonal proteases[29–31] to control protein degradation. The field would benefit from the development of protein-based RNA-regulation platforms with similar properties that enable control of protein production, e.g., for regulation of secreted proteins. Some protein-based RNA-level regulators exist, mainly using translational repressors[32–34] and RNA-cleaving Cas proteins[35–41], but there is still a great need for demonstrated scalability and composability. Also, all of these examples canonically act as only repressor-like "OFF" switches; several RNA-based activator-like "ON" switches have been developed[32,35,42–44], but none of these are composable for circuit engineering. The field currently lacks an RNA-acting activation/repression toolkit that is scalable, robust, modular, and composable to use in place of transcription factors.

Here we develop a post-transcriptional regulation platform that exploits RNA cleavage to control transcript degradation. We term this platform "Programmable Endonucleolytic Scission-Induced Stability Tuning" (PERSIST). We use CRISPR-specific endoribonucleases (endoRNases) as effectors for RNA cleavage-based regulation and adapt a set of nine that are orthogonal in their cleavage to serve as RNA-level "activators" and "repressors". We create RNA-level ON-switch and OFF-switch motifs that can be engineered into the untranslated regions (UTRs) of transcripts and can be modified with a cognate endRNase RNA target sequence to be activated or repressed by specific endoRNases. The endoRNases exhibit up to 300-fold dynamic range as repressors and up to 100-fold dynamic range as activators in the PERSIST platform. When benchmarked against Tet-On, this regulatory modality is less vulnerable to silencing compared to transcriptional regulation because it can make use of vetted constitutive promoters routinely used in gene and cell therapies.

We demonstrate that our endoRNase platform exhibits modularity and composability by creating multi-level cascades, all 16 two-input Boolean logic functions, and a positive feedback motif. A useful feature of this platform is that these endoRNase regulators can function as both activators and repressors simultaneously, which allows for compact engineering of previously complex motifs such as a feed-forward loop and RNA-level bistable switch. The performance and simplicity of the PERSIST platform, combined with its resistance to silencing, will enable the facile construction of genetic circuits with extended functional lifetimes for applications in biomanufacturing, gene therapies, and engineered cell therapies.

## Results

### Development of RNA-level switches that respond to RNA cleavage and resist epigenetic silencing.

We first set out to develop RNA-level switches that enable turning transgene expression on or off through the regulation of transcript degradation. Previous studies have shown that transcript cleavage, for example by miRNA[45] or endoribonucleases[35,46,47], can reduce transgene expression (Fig. 1a, left). Our initial objective was to develop an RNA-level ON switch that activates gene expression in response to programmable transcript cleavage. We designed this motif to have three domains (Fig. 1a, right): (1) RNA degradation signals that elicit rapid degradation of the transcript, (2) a cleavage domain that allows removal of the degradation tags, and (3) a stabilizer that ensures efficient translation and protects the mRNA after removal of the degradation signal. Thus, the transcript is degraded in the absence of a cleavage event and is stabilized post-cleavage.

First, we sought to identify a mechanism that leads to rapid mRNA degradation. A natural short 8bp RNA motif has been recently identified that binds heterogeneous nuclear ribonucleoproteins, which in turn recruit deadenylase complexes, resulting in mRNA degradation in a variety of cell types[48]. We created EYFP reporters that contain varied numbers of repeats (up to 30) of the degradation motif, "wt1", within the 3′ UTR (Supplementary Figs. 1 and 2) and evaluated expression via transient transfection into HEK293FT cells. A reporter containing 30 repeats was efficiently degraded with over 300-fold reduction in fluorescence, while fewer numbers of repeats allowed for variable degrees of repression (Fig. 1b, Supplementary Figs. 3). Notably, while various methods exist to program RNA production, protein production and protein degradation (e.g., with promoter design[49,50], upstream ORFs[51], and destabilization domains[52,53], respectively), repeats of these RNA degradation motifs represent a flexible way to fine-tune mRNA half-life.

Next, we searched for a genetic sequence that would stabilize the transcript and enable efficient translation after the removal of these degradation tags. Importantly, this element should not stabilize the transcript when the degradation tags are still present. An RNA triple helix (triplex) structure from the MALAT1 long-noncoding RNA[54–56] has been used in several synthetic biology studies to stabilize engineered transcripts[35,46,57]. When we inserted the MALAT1 triplex upstream of 30 repeats of the wt1 degradation motif, the expression of the EYFP reporter remained low (Fig. 1c). Then, when we included a mascRNA sequence, which is cleaved naturally by endogenous RNase P, between the triplex and degradation motif, expression was restored to constitutive levels (Fig. 1c). Overall, this configuration exhibits a 166-fold increase in expression relative to the configuration lacking the RNAse P site, which is promising for its adaptation as an RNA-ON-switch motif for transgene regulation.

Based on these encouraging results, we next developed an endoRNAse-responsive ON switch using Csy4. The CRISPR endoRNase Csy4 (PaeCas6f) has been used as a translational repressor in mammalian synthetic biology previously[35,46,47] and here we show that it can be repurposed to activate expression when its cleavage site is placed within the PERSIST-ON motif (Supplementary Figs. 4). It was important to validate the ability of this RNA-ON-switch to resist epigenetic silencing. We chose to

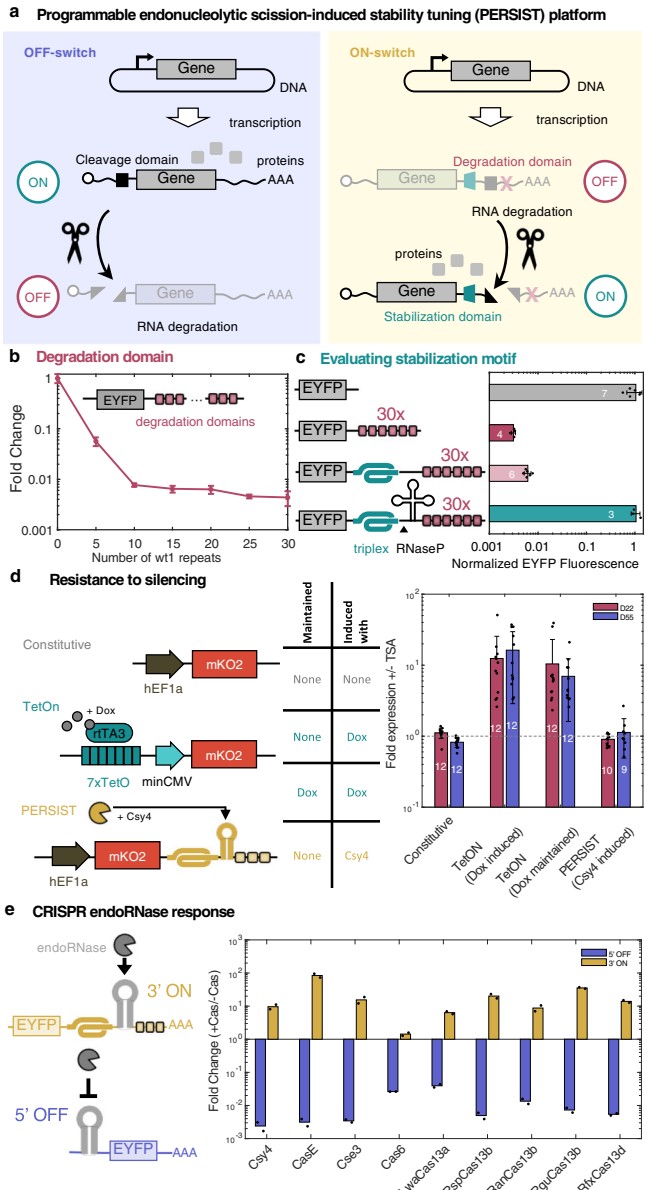

**a** Programmable endonucleolytic scission-induced stability tuning (PERSIST) platform

**b** Degradation domain

**c** Evaluating stabilization motif

**d** Resistance to silencing

**e** CRISPR endoRNase response

**Fig. 1 Engineering PERSIST-OFF- and ON- motifs for regulated gene expression. a** Schematic of RNA-based OFF-switch and ON-switch motif designs that are regulated by RNA cleavage. **b** Gene expression reduction due to appending increasing numbers of degradation motifs. Fold change was calculated as described in Supplementary Fig. 3. $n = 3$ biologically independent samples where each sample represents the evaluation of >1000 transfected cells (HEK293FT). Data are presented as mean ± s.d. **c** Triplex structure rescues gene expression only after removal of degradation motifs by RNase P. Data were analyzed from the total number of samples indicated in the bars where each sample represents the evaluation of >1000 transfected cells (HEK293FT). Data are presented as mean ± s.d. **d** PERSIST shows resistance to epigenetic silencing in clonal populations. Three different constructs (left) each expressing mKO2 (constitutive, Tet-On switch, or PERSIST-ON switch) were genomically integrated and single-cell sorted to evaluate clonal silencing effects (see Supplementary Fig. 5). The table (middle) shows the maintenance and induction conditions of each cell line. 22 and 55 days after sorting 12 clonal lines for each cell type were induced with either 4 μM Dox or transfected with Csy4 both in the presence and absence of TSA and evaluated by flow cytometry two days later. The ratio of ±TSA for mKO2 expression was calculated for each cell line in each sample type where a ratio >1 indicates rescue of response by TSA. The Tet-On system shows increased mKO2 expression in the presence of TSA, which indicates epigenetic silencing effects. The PERSIST switch resists silencing similar to the constitutively expressed reporter. Data were analyzed from the total number of clones indicated in the bars where each sample represents the evaluation of ≥6 CHO-K1 cells. Data are presented as mean ± s.d. **e** CRISPR endoRNases activate the PERSIST-ON motif and repress the PERSIST-OFF motif. Fold change is normalized to the effect of adding endoRNase to a control reporter without an endoRNase recognition site as described in Supplementary Fig. 4. Constructs and endoRNases were optimized (see Supplementary Figs. 6–8; the best variants are shown here). $n = 2$ biologically independent samples where each sample represents the evaluation of >1000 transfected cells (HEK293FT). Data bars are presented as mean.

inhibitor Trichostatin A (TSA) (Supplementary Fig. 5c), which would rescue any loss in response specifically due to silencing via deacetylation. We chose to evaluate an HDAC inhibitor because, as shown by Oyer et al.[60], HDAC activity rather than DNA methylation was mainly responsible for driving epigenetic silencing of the Tet system. As seen in Fig. 1d both the constitutive reporter and the PERSIST-ON-switch respond similarly with and without TSA as indicated by their ± TSA ratio being close to one. This implies that both of these constructs resist HDAC-related epigenetic silencing. Interestingly, rescue with TSA addition was observed for the Tet-On system regardless of whether or not it was maintained in Dox, implying that both cases experience epigenetic silencing. Whether this effect is attributed to the properties of the Tet-On promoter sequence itself or the transcriptional activator, it is clear that the PERSIST-ON-switch under hEf1a promoter avoids these pitfalls and enables long-term robust yet regulatable response. The reliability of the response suggests that our post-transcriptional platform for transgene regulation could prove useful for applications where long-term control is required such as in cell therapies and biomanufacturing.

**A CRISPR endoRNase library for the PERSIST platform**. In order to generate a library of RNA-regulators, we next sought to evaluate whether other endoRNases besides Csy4 could cleave the transcript and activate the ON-switch motif or repress the OFF-switch motif. The Cas6 family that Csy4 belongs to and the Cas13 family of CRISPR-specific endonucleases cleave pre-crRNAs to

benchmark the long-term inducibility of the Csy4-responsive PERSIST-ON-switch against the routinely used Tet-On system[58]. In the Tet-On transcriptional ON-switch, rtTA3 is induced by doxycycline (Dox) to activate the expression of the TRE promoter. We engineered both the Tet-On and PERSIST switches to drive the expression of the fluorescent protein mKO2 and compared it to a control construct encoding constitutive mKO2 expression from a hEf1a promoter (Fig. 1d). All three constructs were integrated into genomic landing pads located in the putative Rosa26 locus of CHO-K1 cells[59] and polyclonal populations showed expected responses (Supplementary Figs. 5a). We then used single-cell sorting to generate 12 cell lines for each of four different conditions tested: (1) constitutive expression, (2) Tet-On switch maintained without Dox until induction, (3) Tet-On switch always maintained in the presence of Dox to simulate constitutive expression, and (4) PERIST ON-switch (Supplementary Fig. 5b). To evaluate response, each of the switches was induced (either by Dox addition or Csy4 transfection) 22 days and 55 days after sorting. We measured mKO2 levels in the presence and absence of the histone deacetylase (HDAC)

produce shorter gRNA sequences used for DNA- or RNA-targeting respectively[61]. Importantly, these endonucleases cleave short, specific (often hairpin-structured) sequences called direct repeats, and orthologs of these endonucleases are thought to cleave their respective direct repeats orthogonally[62]. We hypothesized that these Cas6 and Cas13 families could provide a large number of parts that orthogonally cleave mRNAs when their cognate cleavage domain is engineered into the transcript. We evaluated nine total endonucleases, four from the Cas6 family: Csy4, Cse3 (TthCas6e)[63], CasE (EcoCas6e)[64] and Cas6 (PfuCas6-1)[65] and five from the Cas13 family: LwaCas13a[36], PspCas13b[38,39], PguCas13b[38,39], RanCas13b[38,39], and RfxCas13d[40]. Since Csy4 and all of the Cas13 family endonucleases have previously been used in mammalian cells, only Cse3, CasE, and Cas6 required mammalian codon optimization; we made several other modifications to the Cas13 family endonucleases to improve performance and ensure compatibility with other endonucleases (Supplementary Fig. 6). We observed robust ON responses to most of these endoRNases when their cognate recognition sequences were placed between the triplex and degradation signal in the 3′ UTR-located PERSIST-ON-switch motif (Fig. 1e and Supplementary Figs 4 and 7). Interestingly, for some Cas endoRNases, the triplex was not required for ON-switch functionality (Supplementary Fig 7), likely due to the fact that these endoRNases stay bound to their cleaved product and protect the 3′-end of the transcript as previously shown[66–68]. The same set of endonucleases led to robust OFF responses of up to 300-fold repression when only their target sequences were placed into the transcript's UTRs with larger repression occurring when placed in the 5′ UTR for those tested (Fig. 1e, Supplementary Fig. 8). We also evaluated PERSIST-ON- and OFF-switch dynamics in response to the endoRNase CasE (Supplementary Fig. 9) and verified that the PERSIST-ON-switch responds to CasE in several other relevant cell lines: CHO-K1, HeLa, and Jurkat (Supplementary Fig. 10), suggesting that the native degradation and stabilization mechanisms employed by the PERSIST platform are ubiquitous.

**PERSIST switches demonstrate orthogonality, composability, and modularity**. The repression and activation responses to CRISPR endoRNases in the 5′ UTR OFF-switch and 3′ UTR ON-switch positions respectively suggested that these proteins could be used as RNA-level activators and repressors for programmable gene circuits. To validate PERSIST as a platform that could be used in place of or alongside transcription factor-based regulation of genetic circuits, we next sought to evaluate the orthogonality, composability, and modularity of the Cas-based PERSIST switches as shown below.

We first evaluated the orthogonality of the Cas proteins by testing each Cas protein with every pairwise combination of Cas-responsive PERSIST-OFF and -ON reporters (Supplementary Fig. 11–13). Notably, CasE strongly cleaves the Cse3 recognition hairpin, but a single mutation U5A in the Cse3 recognition motif[63] (Cse3*) renders it cleavable only by Cse3 and not CasE. As seen in Fig. 2a, the observed orthogonality suggests that subsets of this set of proteins could be used within the same circuit. Certain pairs, such as RanCas13b/PguCas13b and CasE/Cse3 (with the wt Case3 recognition site), should be avoided unless the cross-reactivity is beneficial for circuit design.

Next, we evaluated the composability of the PERSIST platform by layering the endonucleases. Here we make use of PEST-tagged endoRNases (Csy4, CasE, and Cse3)(Supplementary Fig. 14) in intermediate layers to promote faster state-switching which is useful for layered circuits. In Fig. 2b we demonstrate a two-stage repression cascade where one endoRNase (CasE) represses

another (Csy4-PEST), that in turn represses a reporter. We show reporter EYFP expression as a function of CasE expression (proportional to mKO2 fluorescence) at three different Csy4:CasE ratios and with no Csy4. In the absence of Csy4, EYFP is strongly expressed (gray line). When varying amounts of Csy4 DNA are added and CasE is expressed at low levels, the reporter is also expressed at low levels. As the level of CasE increases, it represses Csy4, and expression of EYFP is restored. Note that at higher ratios of Csy4 to CasE, CasE is oversaturated and cannot fully repress Csy4, and therefore full fluorescence levels cannot be restored, likely because Csy4 is a slightly stronger repressor than CasE (see Fig. 1e). While other post-transcriptional platforms, namely protease-mediated regulation[29–31], have similarly shown repression of another repressor, none have demonstrated direct activation of a repressor. Therefore, next, we demonstrate a two-stage activation/repression cascade where CasE activates Csy4, which, in turn, represses EYFP (Fig. 2c). In the absence of Csy4, EYFP is expressed strongly (gray line). When varying amounts of Csy4 DNA are added and CasE is expressed at low levels, Csy4 is not yet activated and the reporter remains highly expressed. As the level of CasE expression increases, it activates Csy4 which represses the reporter. Fig. 2d shows auto-positive feedback with endoRNase self-activation achieved by placing its recognition site in the 3′ ON-switch of its own transcript. The level of output fluorescence for the positive feedback is high for all transfection levels and, interestingly, seemed to even slightly surpass constitutive expression levels, perhaps due to increased mRNA stability provided by CasE binding.

We then sought to show that the Cas protein recognition sites provide modularity such that multiple recognition sites can be placed in a single transcript, in the 5′-OFF and/or 3′-ON PERSIST switch locations, and have predictable behavior. We demonstrate this modularity by creating all 16 two-input Boolean logic functions using transcripts targeted by multiple endoRNases (Fig. 3 and Supplementary Fig. 15). Importantly, multiple recognition sites can be combined within the same 5′ OFF or 3′ ON PERSIST motif (see "NOR" and "OR") or within separate motifs on the same transcript (see "NIMPLY"). In fact 10 of the 16 logic gates can be constructed with a single transcript and they all perform well with dynamic ranges approaching two orders of magnitude. Importantly, all logic functions can be accurately identified using the same threshold values (gray region). Overall, these logic gates compare well to previous post-transcriptional platforms as well as DNA-based logic gates in mammalian cells[69–71].

**Dual function enables compact implementation of complex logic**. Unlike transcription factors in DNA logic-based systems, which usually function to either repress or activate transcription, these endoRNases can act as both "activators" and "repressors" by placing their recognition sequence in a PERSIST-ON or PERSIST-OFF motif respectively. There are several transcriptional systems that have devised clever ways of switching between activation and repression[6], but these systems show modest fold change and lack broad composability. Importantly, in the PERSIST platform, since a given endoRNase's activation/repression function is determined by its target's location, endoRNases can exhibit this dual functionality simultaneously. This is shown in Fig. 4a, where both a reporter encoding a Csy4-repressible tagBFP transcript and a reporter encoding a Csy4-activatable EYFP transcript can be acted on at the same time in the same cell by Csy4. As described below, we took advantage of this feature of the PERSIST platform to compactly build several motifs including a feed-forward loop (FFL) and a single-node positive feedback/repression motif, which in turn enabled a more robust three-stage cascade and a two-node bistable switch.

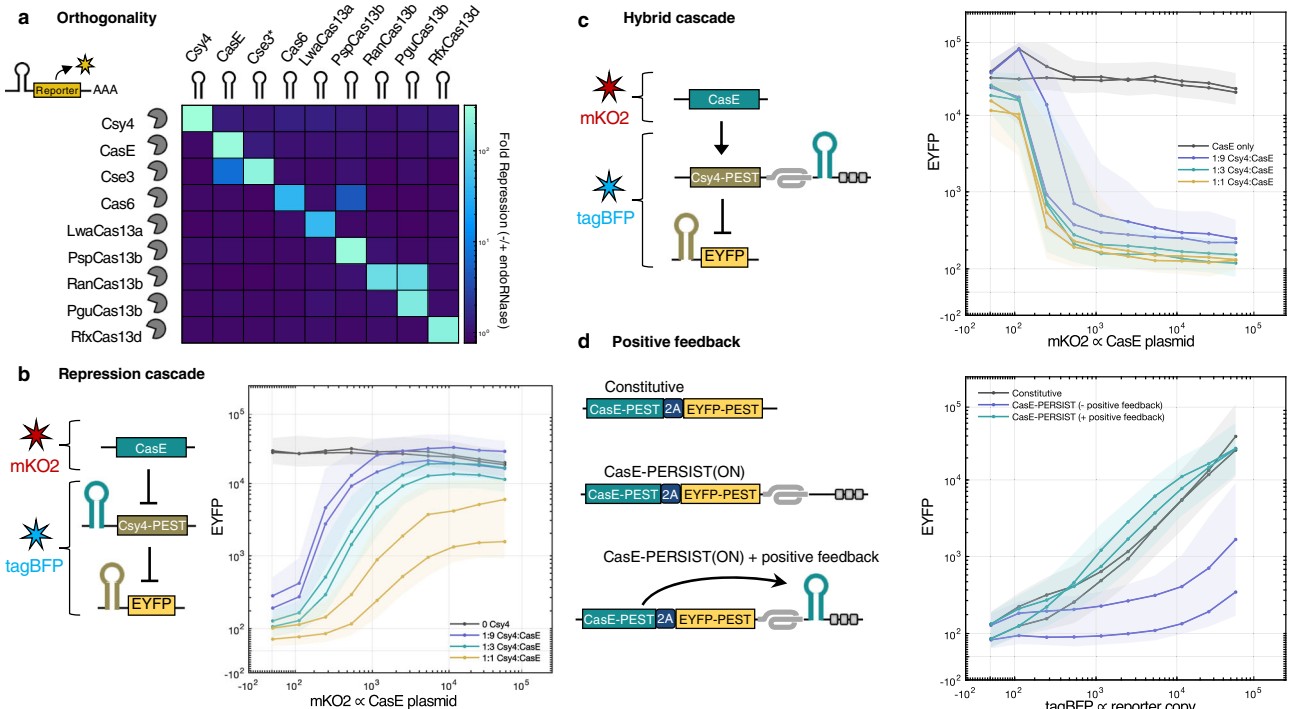

**Fig. 2 PERSIST endoRNase platform is orthogonal and composable. a** Evaluation of Cas endoRNase pairwise orthogonality. The ability of each Cas endoRNase to repress reporters containing each endoRNase recognition site was evaluated by poly-transfection[73]. Fold repression values are indicated by color bar. While some pairs should be avoided, most endoRNases cleave orthogonally (see Supplementary Figs. 11–12). **b** Cas endoRNase two-stage repression cascade. **c** Cas endoRNase two-stage activation/repression cascade. **b, c** For plasmid ratio calculations, CasE plasmid was maintained at 75 ng. CasE plasmid transfection efficiency was tracked by a plasmid encoding constitutive mKO2 (x axis) while Csy4 and EYFP reporter plasmid transfection efficiencies were tracked by a plasmid encoding constitutive tagBFP. Summary values were calculated for tagBFP-positive cells. **d** Cas endoRNase auto-positive feedback. We compared a CasE construct tagged with EYFP alone (gray line), a construct containing the PERSIST-ON motif but no endoRNase target site (blue line), and a construct containing a CasE target site in the PERSIST motif (green line). Construct transfection efficiencies were tracked by a plasmid encoding constitutive tagBFP. **b–d** $n = 2$ biologically independent samples as shown by separate curves where each sample represents the evaluation of >1000 transfected cells (HEK293FT). The shaded region is the maximum interquartile range of all cells in each bin.

The coherent feed-forward loop (cFFL) motif provides redundant regulation, where regulation of a downstream gene via multiple pathways often yields a greater dynamic range. We hypothesized that a cFFL could increase the dynamic range of the PERSIST-ON response to an endoRNase input. In our cFFL, CasE acts as the input and Csy4 as an intermediate node, where both act on the output (Fig. 4b). When CasE is absent, the reporter is both degraded by the ON switch motif and repressed by Csy4, resulting in 10-fold lower background expression than the PERSIST-ON-switch alone. When CasE is present, it directly activates the reporter and indirectly activates it by repressing Csy4. In this cFFL the input endoRNase simultaneously acts as an activator for the reporter and repressor for Csy4, resulting in dynamic ranges of over 1000-fold (Fig. 4b and Supplementary Fig. 16).

Auto-positive feedback is generally considered to have useful properties such as a response with a higher threshold, allowing for better separation between ON and OFF states; when auto-positive feedback then regulates a repressor, the repression response can also have a higher threshold. A dual-function PERSIST device can encode this functionality compactly (Fig. 5a). We constructed such a motif using Csy4 positive feedback and an EYFP OFF-switch reporter containing the Csy4 recognition site in its 5′ UTR (Fig. 5b). In a control experiment, when Csy4 functions just as a repressor, it is expressed stably and represses the reporter even at low levels. In a second control experiment, when Csy4 repressor contains the ON switch degradation tags but is unable to self-activate (the ON switch does not contain a

recognition site), Csy4 transcript is degraded resulting in poor repression of the reporter. When the Csy4 recognition site is placed in the ON switch motif, Csy4 activates itself through a positive feedback loop and recovers its ability to target and repress the reporter.

We further demonstrate the robustness of this motif through the construction of a three-stage repression cascade (Fig. 5c and Supplementary Fig. 17). In a repression-only version of the cascade, there is indeed a drop in signal restoration at stage-2 (Fig. 5c, left), but this cascade still outperforms previous RNA-based repression cascades[34]. When positive feedback is included for stage 1 and 2, a larger percentage of stage-2 cells exhibit "ON" state behavior (Fig. 5c, right). There is, however, a trade-off as a larger percentage of stage three cells exhibit incorrect "ON" behavior when positive feedback is included.

To further demonstrate this motif's utility we sought to create a bistable toggle switch. Notably, it has been shown that DNA-based toggle switch topologies that include positive auto-feedback loops at each node improve bistability compared to those with cross-repression alone[72] and similarly that RNA-based switches relying only on cross-repression do not exhibit bimodality in an uninduced state[34]. Here, our positive feedback + repression motif enables the compact construction of a bistable toggle switch that requires only two nodes encoding both cross-repression and positive feedback. Csy4 and Cse3 both activate themselves as well as repress the other endoRNase (Fig. 5d). We design separate constructs containing fluorescent proteins (mKO2 and EYFP) that can be repressed by each of the endoRNases to track cell

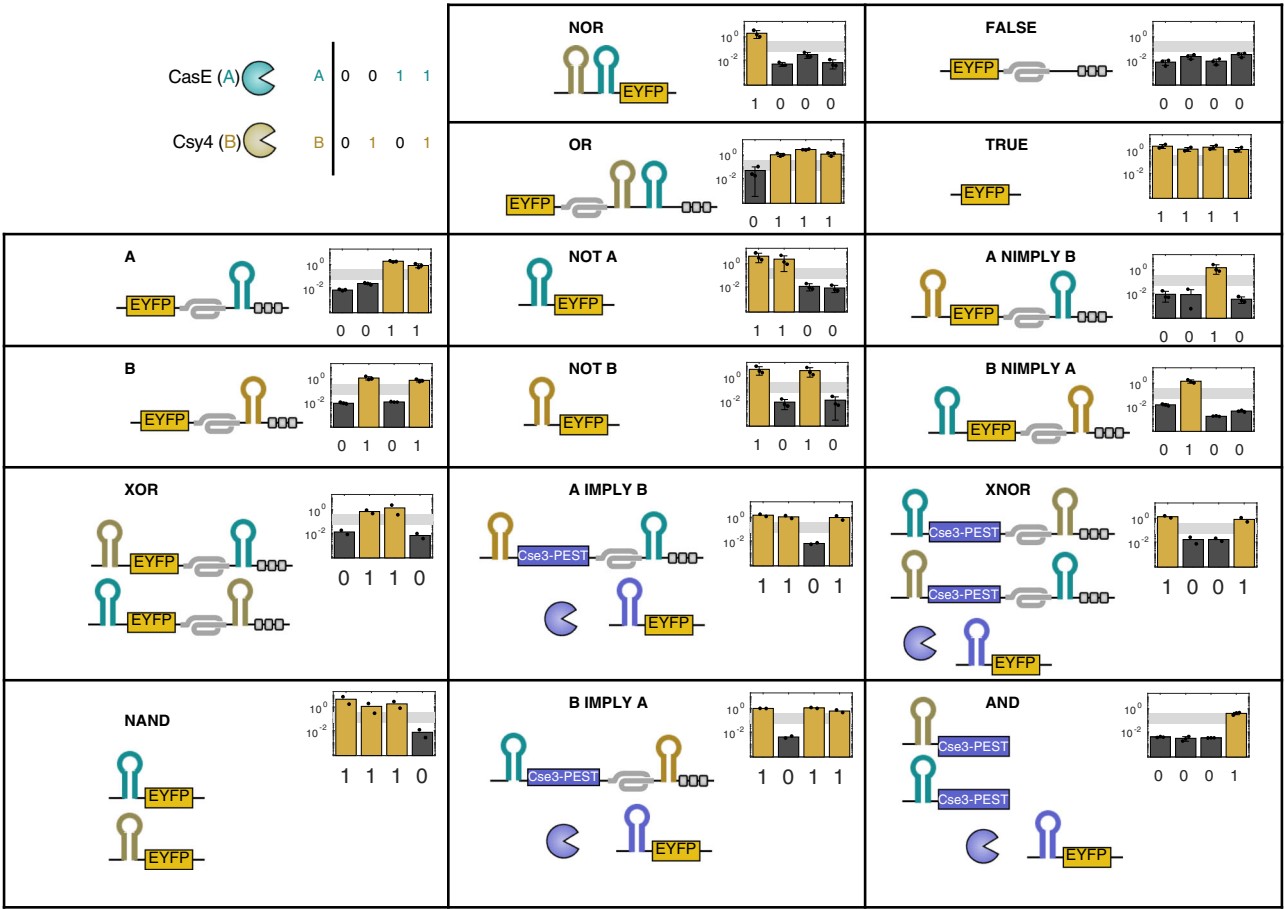

**Fig. 3 PERSIST endoRNase platform is modular.** All 16 two-input Boolean logic functions can be created using Cas endoRNases and the PERSIST-ON- and OFF-switches. Each logic function was constructed using EYFP as output with other constructs included as necessary to confer a designed response to two inputs: Csy4 and CasE. Cse3 endoRNase was used as an intermediate regulator where necessary. Each set of constructs encoding specific logic functions was evaluated without input endoRNases, with CasE only, with Csy4 only, and with both Csy4 and CasE via poly-transfection (Supplementary Fig. 15). The expected Boolean logic output is depicted below each chart. All logic functions can be correctly identified using the same approximately 10-fold separation between ON and OFF values (gray region). $n = 2$ or $n = 3$ biologically independent samples as indicated by the points where each sample represents the evaluation of >1000 transfected cells (HEK293FT). Data bars are presented as mean normalized output.

state. We then evaluated these constructs via transient poly-transfection[73] to allow for many ratios of each node to be sampled (Fig. 5d, right and Supplementary Fig. 18). We achieved thorough sampling of node ratios by creating separate poly-transfection complexes for each node; we tracked the transfection efficiency of the Csy4 node containing the endoRNase plasmid and EYFP output using a constitutive tagBFP reporter while we measured the transfection efficiency of the CasE node containing the endoRNase plasmid and mKO2 output using an iRFP720 reporter. Cells transfected with both nodes (determined by tagBFP and iRFP720 fluorescence) exhibited two major states: a high-mKO2/low-EYFP state and low-mKO2/high-EYFP state. Promisingly, despite the wide range of ratios sampled, 97% of cells were in one state or the other and only 3% of cells expressed none or both fluorescent proteins, implying that this system is indeed bistable.

## Discussion

Here we present a composable, scalable, and programmable RNA activation/repression platform that can be used in place of transcription factors. While other post-transcriptional platforms exist (e.g., protease-mediated regulation of protein degradation), to our knowledge, PERSIST is the first RNA-regulation platform that

has demonstrated both composable activator- and repressor-like regulators. Protease-based regulation platforms[29–31] have garnered much excitement and have advantages, for example, in situations such as mRNA therapy where circuits can be encoded on a single transcript. However, while innovative, the engineering complexities required for creating proteases that regulate not just an output protein but other proteases in more complex circuits hinder their robustness and modularity. For example, none of the protease platforms to date have been able to demonstrate protease-mediated direct activation of another single protease molecule. By separating the effector level (protein) from the level of the regulated substrate (mRNA) we have been able to simplify the design and composition of post-transcriptional regulatory motifs, which may allow these platforms to be used in more complex circuits. Together, both protease-based platforms and our PERSIST platform represent important steps in the synergistic development of post-transcriptional regulation capabilities for use in a wide variety of applications. Importantly, all post-transcriptional platforms enable the use of vetted strong constitutive promoters that resist epigenetic silencing. We show that the PERSIST platform can resist epigenetic silencing for at least 2 months, which can provide important benefits for durable long-term conditional gene expression in cell-based therapies and biomanufacturing.

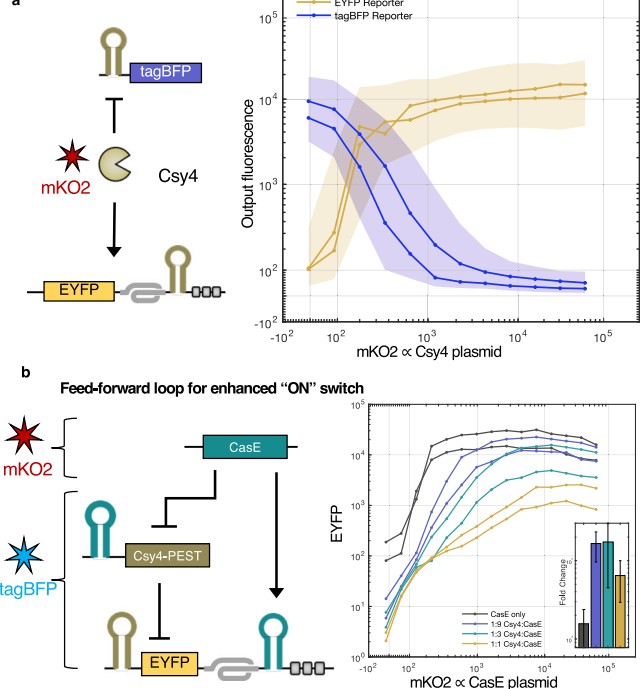

**Fig. 4 Dual function of Cas endoRNases in the PERSIST platform. a** A tagBFP-encoding OFF-switch reporter (blue line) and an EYFP-encoding ON-switch reporter (yellow line) can be acted on simultaneously by Csy4 in the same cell. The ON/OFF-switch reporters and Csy4 plasmid were co-transfected with plasmids encoding constitutive iRFP720 and mKO2 respectively in order to track their corresponding transfection efficiencies. Summary values were calculated for iRFP720-positive cells. **b**, A coherent feed-forward loop with dual function Cas protein improves the response of the PERSIST-ON-switch motif. For plasmid ratio calculations, CasE plasmid was maintained at 75 ng. Transfection efficiency of the EYFP reporter and intermediate Csy4 plasmid was tracked by a plasmid encoding constitutive tagBFP while CasE transfection efficiency was tracked by a plasmid encoding constitutive mKO2. Summary values were calculated for tagBFP-positive cells. Fold change (inset) was calculated by normalizing to the EYFP value at the lowest mKO2 bin (see Supplementary Fig. 16) and evaluated at mKO2 = 10,000. Over 1000-fold dynamic range is achieved for certain Csy:CasE ratios compared to roughly 100-fold dynamic range achieved by the ON-switch alone (gray). **a**, **b** $n = 2$ biologically independent samples where each sample represents the evaluation of >1000 transfected cells (HEK293FT). The shaded region is the maximum interquartile range of all cells in each bin.

Activation of an endoRNase by another endoRNase shown here was enabled by our development of an RNA-level ON switch, which has been a challenge in the field due to the fact that translation is typically "on" by default. While a primary focus of the work described here is endoRNase-based activation, we also explore whether our PERSIST-ON-switch can be directly activated by two other types of RNA cleavers: ribozymes (Supplementary Fig. 19) and miRNAs (Supplementary Fig. 20). Ribozyme-responsive PERSIST-ON switches could be useful for detecting endogenous ribozymes or regulating expression based on small molecules via aptazymes. Furthermore, with additional optimization, single-transcript miRNA-responsive PERSIST-ON switches could serve as genetically encoded reporters or actuators that detect and respond to high levels of miRNA. These so-called "miRNA high sensors" have been difficult to design and optimize in part because they often require a double-inversion step[74] or multiple parts[75]. Therefore, it may be possible to extend the

PERSIST platform for the detection of single-input and multi-input endogenous RNA cleavers such as disease-relevant endoRNases, ribozymes, and miRNAs.

We found that CRISPR endonucleases exhibit ~100-fold dynamic range in repression and >10-fold dynamic range inactivation through the PERSIST-OFF and -ON switches respectively. The fold dynamic range of the ON switch was further improved to >1000 by using an endoRNase-encoded cFFL. Using these endoRNases as RNA-level activators and repressors we demonstrate a range of genetic computations that give us confidence that this system could be used in place of transcription factor-based regulation in scenarios where epigenetic silencing could be problematic. In fact, unlike typical transcription factor operation, the repressor or activator functionality of the endoRNases is determined by the 5′ vs 3′ UTR location of their cognate recognition sites rather than encoded in the protein itself. While transcriptional logic often involves optimization of various repeats of operons with careful placement with respect to minimal promoters to achieve the desired response, programming with the PERSIST platform is straightforward and involves placing the target recognition sites either in the 5′ UTR PERSIST-OFF position or the 3′ UTR PERSIST-ON position. In another work, we have used CasE, one of the most robust endoRNases explored here, to implement an incoherent feed-forward loop that achieves control of gene expression regardless of DNA copy number and resource competition[76]. We also demonstrate that the unique dual functionality of these endoRNases in the PERSIST platform enables the facile engineering of useful motifs. These motifs allow for improved signal restoration and the creation of a compact RNA-level bistable toggle switch. The effective bistability combined with resistance to epigenetic silencing indicates that this switch could serve as a useful tool for studying and modulating cell fate decision making, development of cell type classifiers, and the creation of gene and cell therapies where strict ON or OFF states are required instead of a graded response, for example in cell-based manufacturing or safety switches. The development of small-molecule regulation would enable evaluation of the switch's presumed ability to switch between states and maintain memory after inducer removal and would more generally enable endoRNase-based regulation to be modulated externally in a similar fashion to small-molecule-controllable transcription factors; these longer-term studies are the subject of ongoing work. Finally, here we explored a set of nine endoRNases and showed that they are largely orthogonal in their activity towards their cognate hairpins, which should enable their concurrent use in single circuit topologies. Given a large number of characterized Cas-family proteins with the ability to recognize and cleave specific RNA recognition motifs, PERSIST has the potential to expand beyond these nine proteins, making PERSIST scalable for the construction of large and complex genetic circuits. Taken together, the robustness, scalability, and modularity of the PERSIST platform and its ability to resist silencing promise the programming of sophisticated and reliable behavior in gene and cell therapies.

## Methods

**Plasmid cloning and circuit design.** Hierarchical Golden Gate Assembly was used to assemble all expression constructs from "Level 0" (pL0) parts as published previously[77]. Here we create sub-pL0s that contain the PERSIST-ON-switch stabilizer, cut site, and degradation domains, which can be assembled using BbsI golden gate reactions into a 3′-UTR pL0. We also devise a "Level 2" cloning scheme using SapI golden gate reactions to assemble multiple transcription units onto a single plasmid (Supplementary Fig. 1). Cas6 family expressing plasmids were synthesized by IDT, while Cas13 family genes were cloned from plasmids purchased from addgene: pXR001: EF1a-CasRx-2A-EGFP (Plasmid #109049), pC0046-EF1a-PspCas13b-NES-HIV (Plasmid #103862), pC0045-EF1a-Pgu-Cas13b-NES-HIV (Plasmid #103861), pC0044-EF1a-RanCas13b-NES-mapk (Plasmid #103855), and pC014 - LwCas13a-msfGFP (Plasmid #91902). To ensure

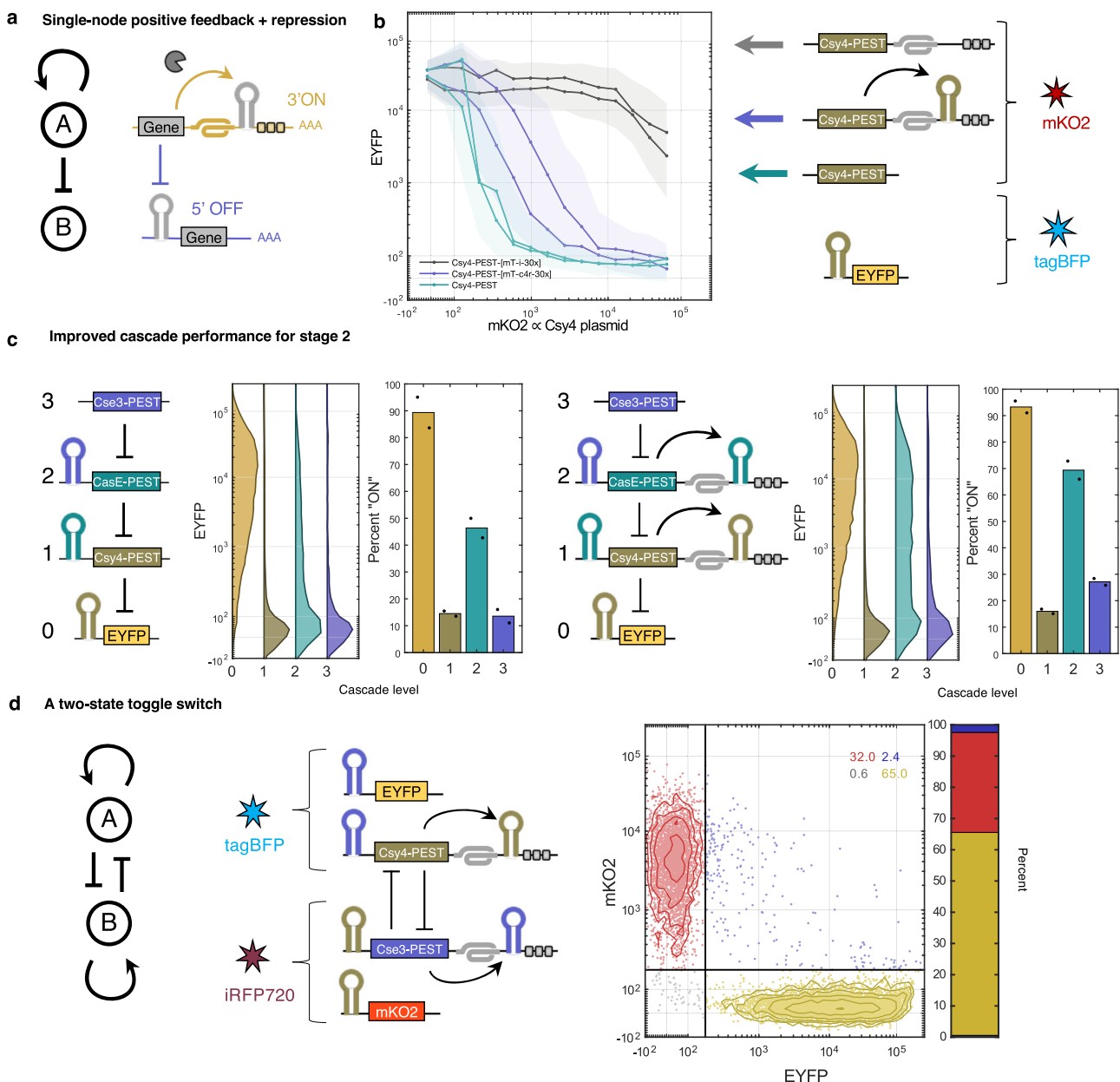

**Fig. 5 PERSIST dual function enables single-element positive feedback + repression motif. a** Schematic of positive feedback + repression motif. **b** Csy4 activates itself through its own PERSIST-ON motif and represses another element. Transfection efficiency of the EYFP reporter was tracked by a plasmid encoding constitutive tagBFP while Csy4-containing plasmid transfection efficiency was tracked by a plasmid encoding constitutive mKO2. Summary values were calculated for tagBFP-positive cells. $n = 2$ biologically independent samples, as shown by separate curves where each sample represents the evaluation of >1000 transfected cells (HEK293FT). The shaded region is the maximum interquartile range of all cells in each bin. **c** The positive feedback + repression motif can be used to improve the performance of the second stage of a three-stage repression cascade. An EYFP reporter tracks the output for subsequent additions of each stage. A larger percentage of stage-2 cells exhibit "ON" behavior with endoRNase positive feedback (right) compared to when repression alone is used (left), with a trade-off existing of an increase in the third stage response. Each bar represents an evaluation of cells within a large range of transfection levels of each stage (Supplementary Fig. 17). $n = 2$ biologically independent samples, where each sample represents the evaluation of >1000 transfected cells (HEK293FT). **d** The positive feedback + repression motif is used to make a genetic bistable switch. mKO2 and EYFP reporter expression was evaluated for iRFP720- and tagBFP-positive cells. Only bistable switch plasmids were used with no additional inputs, yet 97% of cells display expression in only a high-mKO2/low-EYFP (red dots) or high-EYFP/low-mKO2 (yellow dots) state after 48 hours. Few cells expressed high levels of both (blue dots) or none (gray dots). Percentages of cells in each state are shown in the inset and depicted in the bar. $n = 1$ representative sample (HEK293FT).

sufficiently fast circuit dynamics, a PEST tag was fused to any endoRNase within the circuit that is regulated by another endoRNase.

**Cell culture.** HEK293FT (Invitrogen) cells were maintained in Dulbecco's modified Eagle medium (DMEM, Corning) supplemented with 10% FBS (Corning), 1% Penicillin–Streptomycin-L-Glutamine (Corning), and 1% MEM Non-Essential Amino Acids (Gibco). CHO-K1 cells (ATCC) were maintained in Ham's F-12K (Kaighn's) medium supplemented with 10% FBS (Corning), 1% Penicillin–Streptomycin-L-Glutamine (Corning), and 1% MEM Non-Essential Amino Acids (Gibco). Doxycycline Hyclate (Dox, Sigma-Aldrich) was diluted in water at 10 mg/μL, stored at −20 °C, and used at a final concentration of 4 μM for experiments. Trichostatin A (TSA, Sigma-Aldrich) was diluted in DMSO, stored at −20 °C, and used at a final concentration of 100 nM for experiments.

**Transfections**. HEK293FT cell transfections were performed using Lipofectamine 3000 (invitrogen) with a 1.1:1.1:1 Lipofectamine 3000:P3000:DNA ratio in 24-well or 96-well format. Complexes were prepared in Opti-MEM (Gibco). 150,000–200,000 cells or 30,000–50,000 cells were plated on the day of transfection into 24-well or 96-well plates, respectively, in culture media without antibiotics and analyzed by flow cytometry after 48 hours. CHO-K1 cell transfections were performed using Lipofectamine LTX (invitrogen) with a 4:1:1 LTX:Plus Reagent:DNA ratio in 24-well or 96-well format. Complexes were prepared in Opti-MEM (Gibco). Cells were plated on the day of transfection and analyzed by flow cytometry after 48 hours.

**Genomic integration**. Constructs were integrated into CHO-K1 cells that had been previously engineered with a landing pad in their putative Rosa26 locus[59]. For payload integration, cells were transfected with engineered construct and BxBI expressing plasmid in 12-well plate format. After 3 days, media was supplemented with 8 µg/mL puromycin (invigoen) and 10 µg/mL blasticidin (invivogen). Cells were maintained under selection for 2 weeks with fresh media containing antibiotics refreshed every 2 days. After 2 weeks, cell lines were evaluated for response and underwent single-cell sorting.

**Flow cytometry and single-cell sorting**. Cell fluorescence was analyzed with LSR Fortessa flow cytometer, equipped with 405, 488, 561, and 637 nm lasers (BD Biosciences). The following laser and filter combinations were used to evaluate the fluorescent proteins used in transient transfection experiments in this study: tagBFP, 405 nm laser, 450/50 filter; EYFP, 488 nm laser and 515/20 nm filter; mKO2, 561 nm laser, 582/42 filter; iRFP, 640 nm laser, 780/60 filter. See Supplementary Fig. 21 for the sample gating strategy used for all experiments. Cell sorting was performed on a FACSAria cell sorter, equipped with 405, 488, 561, and 640 nm lasers. The following laser and filter combinations were used to evaluate the fluorescent proteins used in single-cell sorting: EBFP, 405 nm laser, 450/40 filter; EYFP, 488 nm laser, 530/30 filter; mKO2, 561 nm laser, 582/15 filter.

**Data analysis**. FACSDiva version 8.0.1 software was used to collect flow cytometry data. Analysis of flow cytometry data were performed in Excel and MATLAB with a custom analysis pipeline. For each data set, a compensation matrix is computed from single color controls to account for spectral bleed-through. After compensation, thresholds were set to evaluate positively transfected cells. For single construct analyses, linear fits were computed between transfection marker (typically tagBFP) and output (typically EYFP) expression levels using the *nlinfit* function (Supplementary Fig. 3). To evaluate fold change in response to endoRNases, slopes from fits were first normalized to those calculated from samples without the cognate endoRNase's recognition site to eliminate non-specific effects associated with just endoRNase addition. For composite devices, points were fitted to a Michaelis–Menten saturation curve (Supplementary Fig. 15). For poly-transfection involving multiple plasmids, data were gated and binned before linear fits or summary statistics were calculated (Supplementary Fig. 4).

**Reporting summary**. Further information on research design is available in the Nature Research Reporting Summary linked to this article.

## Data availability
Representative plasmid sequences have been deposited to Genbank with the accession codes "OM256462 [https://www.ncbi.nlm.nih.gov/nuccore/OM256462]", "OM256463 [https://www.ncbi.nlm.nih.gov/nuccore/OM256463]", "OM256464 [https://www.ncbi.nlm.nih.gov/nuccore/OM256464]", "OM256465 [https://www.ncbi.nlm.nih.gov/nuccore/OM256465]", "OM256466 [https://www.ncbi.nlm.nih.gov/nuccore/OM256466]". All plasmids used in this study are described in this paper and sequence information for parts is provided in Supplementary Data 1. Additional sequence information is available upon reasonable request from the authors. Raw .fcs files are available upon reasonable request. Source data are provided in this paper.

## Code availability
Custom MATLAB scripts are used to process .fcs files, analyze data and generate plots are available upon reasonable request.

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

## Acknowledgements

We thank Kevin Lebo and Ross Jones (MIT) for the discussion, Jin Huh (MIT) for parts, Leonid Gaidukov for providing landing pad cells, Allen Tseng for BxbI plasmid, Emerson Glassey for SapI cloning overhang analysis, and the Koch Flow Cytometry Core for assistance with single-cell sorting. This work was supported by the National Institutes of Health (R01-CA207029: R.W., R01-EB025854: R.W., and R01-CA206218: R.W.), National Science Foundation GRFP (1708200: B.D.).

## Author contributions

B.D. and R.W. conceived the project and designed the experiments. B.D., N.W., E.H., and S.P. performed experiments. B.D. analyzed the data. B.D. and R.W. wrote the manuscript with input from all authors.

## Competing interests

The Massachusetts Institute of Technology has filed patent applications on behalf of the inventors (B.D. and R.W.) for the RNA-ON switch platform described (US Provisional Application No. 16/049,042) as well as on behalf of the inventors (B.D., R.W., and N.W.) for the toggle switch described (US Provisional Application No. 17/122,087). The remaining authors declare no competing interests.
