## [Peer Review File · Nature Communications]

Reviewers' Comments:

Reviewer #1:

Remarks to the Author:

DiAndreth. et al describe a method for post-transcriptional regulation that leverages RNA cleavage to control transcript degradation and describe a series of studies designed to illustrate the potential of this method as a universal control platform. Specifically, this study is motivated by the need to address issues of transcription factor-based expression systems due to epigenetic silencing, which are frequently observed in therapeutically relevant cell types and inhibit cell function over time. The authors suggest that post-transcriptional regulator-based systems would overcome these limitations, allowing the use of constitutive promoters which are resistant to silencing and opening the way to mRNA mediated therapies. While the conclusions are based on a robust set of well-planned and extremely comprehensive experimental studies, the following recommendations are provided to improve this manuscript.

The authors introduce the issue of epigenetic silencing as “hurdles such as epigenetic silencing” that challenge the potential of therapeutic and biomanufacturing applications. Considering that the reader may not be familiar with epigenetic silencing, it is suggested that the authors include a short description to introduce the problem and provide examples of epigenetic silencing and mechanisms.

Figure 2d: The two modes of switching are mediated by different delivery methods (Dox and transfection) and the results of these experiments cannot be directly compared. The experiment should be repeated to include a control plasmid to control for the effects of the transfection procedure, which would allow a more accurate comparison of the two approaches.

No background was given on epigenetic silencing, and no motivation for the use of HDAC inhibitors, as opposed to DNMT inhibitors. It would be useful to explain the reasoning that motivated the selection of an HDAC inhibitor. Such change would also improve readability This addition would also make Figure 1d clearer.

Figure 2a: Orthogonality is only demonstrated for the PERSIST-OFF motif. Context-dependent effects, such as RNA folding, may differentially affect the functioning of orthogonal pairs for the PERSIST-ON motif. Since one of the novelties of this platform is its potential for both activation and repression gene expression, the paper should be revised to include measurements of fold-changes in activation and a demonstration of the orthogonality for the PERSIST-ON motif.

Orthogonality is one of the key features of the platform, yet there is no mention of the orthogonality of the system and associated implications in the discussion section.

Can the authors comment on the extent to which a post-transcriptional regulation-based method such as the platform described in this study is expected to affect the delay between the time of induction of steady state expression of the target gene, possibly in comparison to “traditional” transcriptional regulation based methods and, possibly, post-transcriptional protein regulation-based methods. Since this system is introduced as superior in terms of resistance to epigenetic silencing compared other transcriptional and post-transcriptional regulation-based methods, it would be useful to comment on the dynamic behavior of these systems to provide some context, possibly supported by experimental data which the authors probably have at hand given then have already conducted studies to compare resistance of PERSIST and traditional transcription based systems to epigenetic silencing.

Minor comments:

ABSTRACT:

Line 3

Comma needed after "transcription factor-based regulation" and "based" to

improve readability. The sentence should read as follows:

"However, transcription factor-based regulation, upon which the majority of such applications are based, suffers from complications such as epigenetic silencing, which limits their longevity and reliability."

RESULTS:

Line 65

Syntax: replace "which" with "that"

Line 83:

Remove comma after "Rnase P"

Line 101:

Important: "histone deacetylase Trichostatin A (TSA)" should be "histone deacetylase inhibitor Trichostatin A (TSA)". This omission leads to an opposite reading of the results.

Line 145:

"these set of proteins" should be "these sets of proteins."

Line 146:

Suggested syntax edit "certain pairs, such as RanCas13b:PguCas13b and CasE:Cse3 (with the wt Case3 recognition site), should be avoided."

Reviewer #2:

Remarks to the Author:

The manuscript from DiAndreth et al reports the new gene expression platform that works in RNA-level, called PERSIST. They first designed the mRNA "ON" switches, which has been difficult to develop in this area. PERSIST ON system is consisted of three modules embedded in the 3'-UTR of mRNA; (1) the RNA motif (wt1) that enhances mRNA degradation, (2) the cleaving site for separate the downstream wt1, and (3) the stabilization motif to maintain mRNA after wt1 scission. To induce the separation of wt1, the authors mainly used Cas proteins which have endoribonuclease activities. The performance of PERSIST with Cas proteins was incredibly strong. Additionally, they demonstrated and verified the OFF switches by inserting a cleavage site in 5'-UTR, the orthogonality between 9 Cas proteins, and various synthetic gene circuits including positive feedback and a feed-forward loop that worked at the RNA level. They also showed that PERSIST-ON system integrated into the genome was resistant to epigenetic silencing. The main concept of this study is based on the paper from Borchardt et al (2015), however, the authors newly developed PERSIST, which is renewed as a more versatile and modular platform in the development of post-transcriptional regulation. Thus, I believe this work is important and worth publishing in Nature Communications. I recommend addressing the following concerns before publication.

Major points

1. Almost all of the experiments were performed with 2 biological replicates. In general, these experiments should be performed at least 3 biological replicates.

2. The authors mentioned their PERSIST platform could prove useful tools when long-term control is required, based on the data of Figure 1d and Sup. Fig 5. I have several concerns of PERSIST for this purpose to avoid epigenomic silencing. First, it was unclear for me that PERSIST can generally avoid epigenomic silencing and overcome the issues for TetON silencing. If the promoter region of PERSIST plasmid described in Sup. Fig.5a was mutated and silenced, it should affect the performance of gene activation by Csy4. They used different promoters between two systems (TetON: minCMV, PERSIST/constitutive: hEIF-1a). So, it is difficult to compare the performance of

two systems directly. In other words, the difference of the performance between TetON and PERSIST is simply attributed to the difference of silencing effect between the regions of two promoters? The effect of gene silencing may also be dependent on the cell line. Moreover, although PERSIST seems not to be affected by TSA, the ON/OFF ratio of mKO2 seems to be lower than Tet-ON system as shown in Sup. Fig 5. Especially, the ON state of Tet-ON system had always shown higher mKO2 expression than PERSIST. In addition, in a similar to Tet-ON system, PERSIST may be affected by epigenetic silencing that could not be reset by TSA. Again, what is the advantage of PERSIST compared with Tet-ON system? Are there any additional advantages that the authors can be shown related to long-term control?

3. Related to the author's claim in p11 line 256, I agree that aptazyme-based PERSIST is quite interesting. Can the authors show some advantage of this compared with the previously developed mRNA ON switches in which aptazyme is directly embedded in the 3'-UTR? In addition, considering miRNA-responsive PERSIST data, it seems that this platform is required a strong scission effect to eliminate wt1 motif. Can the authors also demonstrate PERSIST-ON with small molecule-responsive aptazyme? Further discussion with addressing these points may help readers to design and optimize PERSIST dependent on the purpose.

4. The detail of the method to calculate the fold-change should be described in the material and methods section. Was this calculation newly established in this work? If not, please cite the reference. Estimated fold-change values seem to be high compared with simple mean/median/mode-based calculation.

5. Related to figure 3, the authors determined the output cut-off (0.08). Please explain the reason why.

6. The authors often uses PEST-tag. Please explain the reason for each experimental setup. For example, in Figure2b, in repression cascade, CasE was not fused with PEST, but in Fig.5c, all Cas construct was fused with PEST. Similarly, they confirmed the PEST-tagged effect in supplementary figure 12 and showed the improvement of the ON-switch rescue level in Csy4 (except for CasE and Cse3). These PEST-tagged constructs were used in the following experiments (supplementary figure 15, 16). However, I did not know the reasons why PEST-tagged CasE and Cse3 were used.

7. Positive feedback shown in Fig.2d is very interesting. But why this CasE-PERSIST-ON + positive feedback expresses more EYFP compared to constitutive one? Legend c should be "d". Also, the volume of plasmid (75ng) is described in legends but the information of cell number and well should also be important to understand the experimental conditions.

Minor points

1. Some figure legends are not corresponding to their figures correctly. e.g. the text mentioned Figure 2e, but there is no Figure 2e.

2. Please check the references carefully. There are some unlikable citations. e.g. Ref 34 and Ref 57 seem the same paper.

3. In page10 line 238-239, the authors mentioned that "PERSIST is the first RNA regulation platform consisting of both composable activator- and repressor-like regulators." There are reports that have already suggested the dual function(ON and OFF) in RNA-level. For example, Endo et al (PMID: 23999119) showed ON switch and OFF switch could respond to the same trigger protein. In addition, U1A-responsive ON (PMID: 25282610) and OFF switches (PMID: 28525643) have also reported.

4. The detail of the transfection condition is unclear. Please describe each experiment condition.

5. The sequence of wt1 used in this study should be described.

6. Related to Sup figure 13, I assume some linear fit (A IMPLY B, B IMPLY A, XNOR) may unfit the plots. The authors can recheck the data?

7. Single-node positive feedback + repression:

The authors did not describe "Figure 5b". The authors should add "Figure 5b" in a sentence.

8. There are some typo. Please recheck and correct them

Ex. page 4 line 75 : ORFS=>ORFs, page 13 line 303 : -20C => -20°C

9. In Fig. 1e, why the performance of Cas6 is weak compared with others?

Overall Response:

We thank all of the reviewers for their insightful questions, comments, and suggestions. We have carefully incorporated these into our revised manuscript. Within the manuscript any modifications or additions are highlighted in blue text. Our point-by-point responses to reviewer remarks are also indicated in blue text.

REVIEWER COMMENTS

Reviewer #1 (Remarks to the Author):

DiAndreth. et al describe a method for post-transcriptional regulation that leverages RNA cleavage to control transcript degradation and describe a series of studies designed to illustrate the potential of this method as a universal control platform. Specifically, this study is motivated by the need to address issues of transcription factor-based expression systems due to epigenetic silencing, which are frequently observed in therapeutically relevant cell types and inhibit cell function over time. The authors suggest that post-transcriptional regulator-based systems would overcome these limitations, allowing the use of constitutive promoters which are resistant to silencing and opening the way to mRNA mediated therapies. While the conclusions are based on a robust set of well-planned and extremely comprehensive experimental studies, the following recommendations are provided to improve this manuscript.

The authors introduce the issue of epigenetic silencing as “hurdles such as epigenetic silencing” that challenge the potential of therapeutic and biomanufacturing applications. Considering that the reader may not be familiar with epigenetic silencing, it is suggested that the authors include a short description to introduce the problem and provide examples of epigenetic silencing and mechanisms.

We thank the reviewers for this insight. We have added the following sentence to provide background in the introductory section:

“Epigenetic silencing is the prevention of gene expression typically regulated by DNA methylation and chromatin remodeling (e.g. by histone modifications).”

Figure 2d: The two modes of switching are mediated by different delivery methods (Dox and transfection) and the results of these experiments cannot be directly compared. The experiment should be repeated to include a control plasmid to control for the effects of the transfection procedure, which would allow a more accurate comparison of the two approaches.

This is an important comment. While we did consider the approach suggested here, our goal was to keep the current TetOn system as close as possible to its typically-used format to serve as a benchmark. Unfortunately, the development of small-molecule regulation for the PERSIST platform is outside the scope of this manuscript, so we moved forward utilizing standard practice for each method: dox addition for TetOn and Csy4 transfection for PERSIST. While we believe a control where plasmid is transfected could be beneficial, it is unlikely to affect epigenetic silencing and we believe a repeat of this experiment to include this control is unnecessary.

No background was given on epigenetic silencing, and no motivation for the use of HDAC inhibitors, as opposed to DNMT inhibitors. It would be useful to explain the reasoning that motivated the

selection of an HDAC inhibitor. Such change would also improve readability This addition would also make Figure 1d clearer.

We appreciate this suggestion and have added additional details to the manuscript for the choice of HDAC inhibitor:

“We chose to evaluate an HDAC inhibitor because, as shown by Oyer et al., HDAC activity rather than DNA methylation was mainly responsible for driving epigenetic silencing of the Tet system.”

Figure 2a: Orthogonality is only demonstrated for the PERSIST-OFF motif. Context-dependent effects, such as RNA folding, may differentially affect the functioning of orthogonal pairs for the PERSIST-ON motif. Since one of the novelties of this platform is its potential for both activation and repression gene expression, the paper should be revised to include measurements of fold-changes in activation and a demonstration of the orthogonality for the PERSIST-ON motif.

This is a nice suggestion and we have performed the experiment as requested, which is now included as Supplementary Figure 13.

Orthogonality is one of the key features of the platform, yet there is no mention of the orthogonality of the system and associated implications in the discussion section.

We thank the reviewer for noticing this oversight and have included a new discussion of orthogonality in the discussion section (some of which was relocated from the results section):

“Finally, here we explored a set of nine endoRNases and showed that they are largely orthogonal in their activity towards their cognate hairpins, which should enable their concurrent use in single circuit topologies. Given the large number of characterized Cas-family proteins with the ability to recognize and cleave specific RNA recognition motifs, PERSIST has the potential to expand beyond these nine proteins, making PERSIST scalable towards the construction of large and complex genetic circuits.”

Can the authors comment on the extent to which a post-transcriptional regulation-based method such as the platform described in this study is expected to affect the delay between the time of induction of steady state expression of the target gene, possibly in comparison to “traditional” transcriptional regulation based methods and, possibly, post-transcriptional protein regulation-based methods. Since this system is introduced as superior in terms of resistance to epigenetic silencing compared other transcriptional and post-transcriptional regulation-based methods, it would be useful to comment on the dynamic behavior of these systems to provide some context, possibly supported by experimental data which the authors probably have at hand given then have already conducted studies to compare resistance of PERSIST and traditional transcription based systems to epigenetic silencing.

We thank the reviewer for this thought-provoking comment. It is quite possible that post-transcriptional systems like PERSIST have faster switching dynamics than transcriptional regulation systems because only translation is required rather than both transcription and translation. We took a first step towards understanding the system dynamics. The new experimental data is now included as Figure 9 in the Supplementary Information section, where we evaluated PERSIST ON and OFF switch response time to transfected CasE addition.

Minor comments:

ABSTRACT:

Line 3

Comma needed after "transcription factor-based regulation" and "based" to improve readability. The sentence should read as follows:

"However, transcription factor-based regulation, upon which the majority of such applications are based, suffers from complications such as epigenetic silencing, which limits their longevity and reliability."

Thank you—it has been addressed.

RESULTS:

Line 65

Syntax: replace "which" with "that"

Thank you—it has been addressed.

Line 83:

Remove comma after "Rnase P"

Thank you for the comment, however "which is cleaved naturally by endogenous RNase P" is the clause that needs isolation and commas on either side.

Line 101:

Important: "histone deacetylase Trichostatin A (TSA)" should be "histone deacetylase inhibitor Trichostatin A (TSA)". This omission leads to an opposite reading of the results.

Thank you—it has been addressed.

Line 145:

"these set of proteins" should be "these sets of proteins."

Thank you—it has been changed to "this set of proteins"

Line 146:

Suggested syntax edit "certain pairs, such as RanCas13b:PguCas13b and CasE:Cse3 (with the wt Case3 recognition site), should be avoided."

Thank you—it has been addressed.

Reviewer #2 (Remarks to the Author):

The manuscript from DiAndreth et al reports the new gene expression platform that works in RNA-level, called PERSIST. They first designed the mRNA “ON” switches, which has been difficult to develop in this area. PERSIST ON system is consisted of three modules embedded in the 3'-UTR of mRNA; (1) the RNA motif (wt1) that enhances mRNA degradation, (2) the cleaving site for separate the downstream wt1, and (3) the stabilization motif to maintain mRNA after wt1 scission. To induce the separation of wt1, the authors mainly used Cas proteins which have endoribonuclease activities. The performance of PERSIST with Cas proteins was incredibly strong. Additionally, they demonstrated and verified the OFF switches by inserting a cleavage site in 5'-UTR, the orthogonality between 9 Cas proteins, and various synthetic gene circuits including positive feedback and a feed-forward loop that worked at the RNA level. They also showed that PERSIST-ON system integrated into the genome was resistant to epigenetic silencing. The main concept of this study is based on the paper from Borchardt et al (2015), however, the authors newly developed PERSIST, which is renewed as a more versatile and modular platform in the development of post-transcriptional regulation. Thus, I believe this work is important and worth publishing in Nature Communications. I recommend addressing the following concerns before publication.

Major points

1. Almost all of the experiments were performed with 2 biological replicates. In general, these experiments should be performed at least 3 biological replicates.

We thank the reviewer for this comment. Unfortunately, repeating each experiment for a third time would require 664 wells of experimental conditions, 103 of which would be maintained over a two-month time scale. The sheer number of simultaneous experiments that we already performed appears to be in line with the rigor recognized by the community to understand the general trends of such measured devices. In the past month alone, Nature Communications has published several studies in which two biological replicates were sufficient and a third biological replicate would be cost- and time-prohibitive:

- *A small molecule produced by Lactobacillus species blocks Candida albicans filamentation by inhibiting a DYRK1-family kinase* (DOI: <https://doi.org/10.1038/s41467-021-26390-w>)
- *Association of snR190 snoRNA chaperone with early pre-60S particles is regulated by the RNA helicase Dbp7 in yeast* (DOI: <https://doi.org/10.1038/s41467-021-26207-w>)
- *Regulation of plant phototropic growth by NPH3/RPT2-like substrate phosphorylation and 14-3-3 binding* (DOI: <https://doi.org/10.1038/s41467-021-26333-5>)
- *Stage-resolved Hi-C analyses reveal meiotic chromosome organizational features influencing homolog alignment* (DOI: <https://doi.org/10.1038/s41467-021-26033-0>)
- *RN7SK small nuclear RNA controls bidirectional transcription of highly expressed gene pairs in skin* (DOI: <https://doi.org/10.1038/s41467-021-26083-4>)
- *Next generation of tumor-activating type I IFN enhances anti-tumor immune responses to overcome therapy resistance* (DOI: <https://doi.org/10.1038/s41467-021-26112-2>)
- *Thiocysteine lyases as polyketide synthase domains installing hydropersulfide into natural products and a hydropersulfide methyltransferase* (DOI: <https://doi.org/10.1038/s41467-021-25798-8>).

We also draw attention to our decision to analyze data via curve fitting rather than calculating statistics at any one particular point (wherever possible), which increases the robustness of our data against outliers and misleading results.

2. The authors mentioned their PERSIST platform could prove useful tools when long-term control is required, based on the data of Figure 1d and Sup. Fig 5. I have several concerns of PERSIST for this purpose to avoid epigenomic silencing. First, it was unclear for me that PERSIST can generally avoid epigenomic silencing and overcome the issues for TetON silencing. If the promoter region of PERSIST plasmid described in Sup. Fig.5a was mutated and silenced, it should affect the performance of gene activation by Csy4. They used different promoters between two systems (TetON: minCMV, PERSIST/constitutive: hEIF-1a). So, it is difficult to compare the performance of two systems directly. In other words, the difference of the performance between TetON and PERSIST is simply attributed to the difference of silencing effect between the regions of two promoters? The effect of gene silencing may also be dependent on the cell line. Moreover, although PERSIST seems not to be affected by TSA, the ON/OFF ratio of mKO2 seems to be lower than Tet-ON system as shown in Sup. Fig 5. Especially, the ON state of Tet-ON system had always shown higher mKO2 expression than PERSIST. In addition, in a similar to Tet-ON system, PERSIST may be affected by epigenetic silencing that could not be reset by TSA. Again, what is the advantage of PERSIST compared with Tet-ON system? Are there any additional advantages that the authors can be shown related to long-term control?

The reviewer brings up some important points here and we thank them for their comments.

1. In regards to their first point on promoter silencing, as mentioned in our manuscript, we believe that the promoter may be a contributing factor in silencing the TetOn system. In the Results section, we find that continuous expression of TetOn via continuous DOX addition does not rescue silencing resistance and postulate that this result could be due to several factors: (1) “properties of the Tet-On promoter sequence itself” or (2) “the transcriptional activator”. We further say that “the PERSIST ON-switch under hE1a promoter avoids these pitfalls and enables long-term robust yet regulatable response.” Similarly in our introduction we mention that, “when benchmarked against TetOn, [PERSIST] is less vulnerable to silencing compared to transcriptional regulation **because it can make use of vetted constitutive promoters** routinely used in gene and cell therapies.” In other words, the community has already vetted over the course of many years several constitutive promoters such as hEF1a that provide a clear path to obtain expression that is resistant to silencing, but it is not readily apparent that any widely used promoters that can be turned ON and OFF transcriptionally are able to resist silencing in the same manner. By moving the regulation to the RNA level with the PERSIST platform, we can now use these widely used and reliable promoters. It is also impossible to compare transcriptional systems like TetOn and PERSIST using the same promoter. Overall, it is indeed possible (and, as we argue, likely) that the difference in silencing is due to the difference between regions of the two promoters, a postulation that we frequently bring up throughout the manuscript.
2. To the reviewer’s second point concerning mKO2 expression, the Csy4 activation of around tenfold is not surprising here as it is consistent with transient expression data shown in Figure 1e (about 10x for Csy4 activation; close to 100x for CasE activation). Because the reporters for TetOn and PERSIST-ON with Csy4 are regulated differently, we should not expect their absolute ON state expression level to be the same. While TetOn tends to have a larger dynamic range for activation, we believe that the 100x dynamic activation range of PERSIST-ON switches (e.g. using CasE) will prove useful for regulatory networks, especially since we show that this activation range can be further increased to 1000x by using a cFFL

motif (see Figure 4b). Overall, the lower level of mKO2 observed for PERSIST-ON switch in comparison to TetOn results from the properties of Csy4 as an activator and should not be attributed to silencing because it remains consistent over time.

3. Finally, we address the reviewers' third point regarding epigenetic silencing in response to TSA in in the response to reviewer 1 above. In addition, we have also made the following edits to the manuscript to clarify that the reduction in silencing we see is specifically due to reduced histone deacetylation:

“We measured mKO2 levels in the presence and absence of the histone deacetylase (HDAC) inhibitor Trichostatin A (TSA) (Supplementary Figure 5c), which would rescue any loss in response specifically due to silencing via deacetylation.”

“This implies that both of these constructs resist HDAC-related epigenetic silencing.”

3. Related to the author's claim in p11 line 256, I agree that aptazyme-based PERSIST is quite interesting. Can the authors show some advantage of this compared with the previously developed mRNA ON switches in which aptazyme is directly embedded in the 3'-UTR? In addition, considering miRNA-responsive PERSIST data, it seems that this platform is required a strong scission effect to eliminate wt1 motif. Can the authors also demonstrate PERSIST-ON with small molecule-responsive aptazyme? Further discussion with addressing these points may help readers to design and optimize PERSIST dependent on the purpose.

We appreciate that this reviewer is intrigued by the ideas we suggested in the discussion to extend the PERSIST platform further into “sensing” applications. However, as the main focus of this manuscript is the development of an endoRNase-based regulation platform, we believe demonstrating the suggestions that we detail in the discussion section to be outside the scope of this work.

4. The detail of the method to calculate the fold-change should be described in the material and methods section. Was this calculation newly established in this work? If not, please cite the reference. Estimated fold-change values seem to be high compared with simple mean/median/mode-based calculation.

We thank the reviewer for their attention to our analysis methods, and appreciate the chance to elaborate on our methodology. The process by which we fit cotransfection and polytransfection data to appropriate model functions is described in Supplementary Figures 3 and 4, respectively. Our method of calculating fold change is enabled by the recent development of polytransfection methods, and we believe it shows a clear advantage in representing the entire spectrum of the response curve since it avoids bias inherent in choosing one (possibly not representative) transfection bin or ratio between enzyme and reporter. Indeed, simply reporting mean or median fold change at the optimal ratio between endoRNase and reporter plasmid would have resulted for us in significantly higher fold change for many experimental conditions. For example, reporting median fold change in the optimal bin for Csy4 repression of reporter would allow us to claim 541-fold repression by Csy4, and reporting mean fold change in the optimal bin would allow us to claim 321-fold repression. Instead, we report 300-fold repression (Figure 1e) based on our curve fitting that does not constrain our analysis to a single ratio of endoRNase to reporter, but instead allows us to report statistics reflective of a wide range of ratios across the entire polytransfection space.

In order to clarify our analysis methods, we have added the following sentence to the caption of Figure 1:

“Fold changes were calculated as described in Supplementary Figure 3.”

5. Related to figure 3, the authors determined the output cut-off (0.08). Please explain the reason why.

We thank the reviewer for this question. In response to this comment, rather than using a single cutoff, we now depict the figure with the range of values that separate ON from OFF for all logic functions evaluated in Figure 3.

6. The authors often uses PEST-tag. Please explain the reason for each experimental setup. For example, in Figure2b, in repression cascade, CasE was not fused with PEST, but in Fig.5c, all Cas construct was fused with PEST. Similarly, they confirmed the PEST-tagged effect in supplementary figure 12 and showed the improvement of the ON-switch rescue level in Csy4 (except for CasE and Cse3). These PEST-tagged constructs were used in the following experiments (supplementary figure 15, 16). However, I did not know the reasons why PEST-tagged CasE and Cse3 were used.

We thank the reviewer for bringing this concern to our attention. As with previous works (Haynes *et al. ACS Synth. Biol.* 2012, Moore *et al Nucleic Acids Research* 2015, Fukuda *et al ACS Synth. Biol.* 2017) we used a PEST tag to increase the rate of degradation of the proteins which would lead to a faster steady-state response (something that is important to achieve when using transient transfections with limited windows of reliable data). We therefore used a PEST tag for any endoRNase component that was an internal node within the circuit design (i.e. is regulated by another endoRNase). We now include a sentence in the Methods section that describes our choice of whether or not to use PEST fusion to endoRNases in our circuit:

“To ensure sufficiently fast circuit dynamics, a PEST tag was fused to any endoRNase within the circuit that is regulated by another endoRNase.”

7. Positive feedback shown in Fig.2d is very interesting. But why this CasE-PERSIST-ON + positive feedback expresses more EYFP compared to constitutive one? Legend c should be “d”. Also, the volume of plasmid (75ng) is described in legends but the information of cell number and well should also be important to understand the experimental conditions.

We thank the reviewer for bringing up this point on the high level of expression of the positive feedback motif. We also found it very interesting that the transcript with positive feedback appeared to express slightly higher than the constitutive version and have added the following text to the manuscript:

“The level of output fluorescence for the positive feedback is high for all transfection levels and, interestingly, seemed to even slightly surpass constitutive expression levels, perhaps due to increased mRNA stability provided by CasE binding.”

We also thank the reviewer for noticing our oversight in legend labelling and in describing the cell number. We have added the following to the Methods section:

“150,000-200,000 cells or 30,000-50,000 cells were plated on the day of transfection into 24-well or 96-well plates respectively in culture media without antibiotics and analyzed by flow cytometry after 48 hours.”

Minor points

1. Some figure legends are not corresponding to their figures correctly. e.g. the text mentioned Figure 2e, but there is no Figure 2e.

Thank you—it has been addressed.

2. Please check the references carefully. There are some unlikable citations. e.g. Ref 34 and Ref 57 seem the same paper.

Thank you for bringing this to our attention. We have checked the references carefully and addressed this concern.

3. In page 10 line 238-239, the authors mentioned that “PERSIST is the first RNA regulation platform consisting of both composable activator- and repressor-like regulators.” There are reports that have already suggested the dual function (ON and OFF) in RNA-level. For example, Endo et al (PMID: 23999119) showed ON switch and OFF switch could respond to the same trigger protein. In addition, U1A-responsive ON (PMID: 25282610) and OFF switches (PMID: 28525643) have also reported.

This comment was helpful for us to know that the novelty claim is unclear. All of the examples that the reviewers suggest have not demonstrated composability. To make this claim more clear we have altered the sentence slightly:

“While other post-transcriptional platforms exist (e.g. protease-mediated regulation of protein degradation), to our knowledge, PERSIST is the first RNA regulation platform consisting of both demonstrated composable activator- and repressor-like regulators.”

4. The detail of the transfection condition is unclear. Please describe each experiment condition.

We thank the reviewer for this comment and believe it is addressed by the addition to our methods section in response to Major Point 7.

5. The sequence of wt1 used in this study should be described.

We have now included a file with our submission that details the sequences used in this study including wt1.

6. Related to Sup figure 13, I assume some linear fit (A IMPLY B, B IMPLY A, XNOR) may unfit the plots. The authors can recheck the data?

We appreciate the reviewer comment here and agree that our chosen function (linear) did not fit the data. We have, instead, used the Michaelis-Menten function for this figure which behaves much more favorably and is updated in Supplementary Figure 15. We also updated Figure 3 to reflect this new analysis.

7. Single-node positive feedback + repression:

The authors did not describe “Figure 5b”. The authors should add “Figure 5b” in a sentence.

We thank the reviewer for catching this omission and have now included a reference to Figure 5b in the text:

“We constructed such a motif using Csy4 positive feedback and an EYFP OFF-switch reporter containing the Csy4 recognition site in its 5' UTR (Figure 5b).”

8. There are some typo. Please recheck and correct them

Ex. page 4 line 75 : ORFS=>ORFs, page 13 line 303 : -20C => -20°C

Thank you, all have been addressed.

9. In Fig. 1e, why the performance of Cas6 is weak compared with others?

Thank you for this question. Understanding the variation in performance across the Cas proteins is an interesting subject; however, was outside the scope of this manuscript. We hope that the range of activities across the library will actually prove beneficial for engineering various circuit applications and was not something we sought to optimize.

Reviewers' Comments:

Reviewer #1:

Remarks to the Author:

the authors addressed the feedback of this reviewer

Reviewer #2:

Remarks to the Author:

The authors have largely responded to reviewer's comments. The manuscript is ready for publication. Related to my major point 4, does Supplementary Figure 3 show the way of liner fit calculation, and does Supplementary Figure 4 show the way of fold changes calculation? It may be helpful to describe what kind of summary statistics were used for calculation the fold changes in Data Analysis section.

Overall Response:

We thank the reviewers for their helpful comments. We have incorporated the remaining feedback into our revised manuscript and made the modifications in blue text. Our point-by-point responses to reviewer remarks are also indicated in blue text below

REVIEWER COMMENTS

Reviewer #1 (Remarks to the Author):

the authors addressed the feedback of this reviewer

We thank this reviewer.

Reviewer #2 (Remarks to the Author):

The authors have largely responded to reviewer's comments. The manuscript is ready for publication. Related to my major point 4, does Supplementary Figure 3 show the way of liner fit calculation, and does Supplementary Figure 4 show the way of fold changes calculation? It may be helpful to describe what kind of summary statistics were used for calculation the fold changes in Data Analysis section.

We thank the reviewer for this question and identifying our need to elaborate on our methods. Fitting is demonstrated in both Supplementary Figure 3 and in Supplementary Figure 4, while Supplementary Figure 4 shows our method of achieving normalized values used for fold change calculations. To clarify we have added the following text to our Data Analysis section:

“To evaluate fold change in response to endoRNases, slopes from fits were first normalized to those calculated from samples without the cognate endoRNase's recognition site to eliminate non-specific effects associated with just endoRNase addition.”